# Therapeutic Potential of (−)-Agelamide D, a Diterpene Alkaloid from the Marine Sponge *Agelas* sp., as a Natural Radiosensitizer in Hepatocellular Carcinoma Models

**DOI:** 10.3390/md18100500

**Published:** 2020-09-29

**Authors:** Changhoon Choi, Yeonwoo Cho, Arang Son, Sung-Won Shin, Yeon-Ju Lee, Hee Chul Park

**Affiliations:** 1Department of Radiation Oncology, Samsung Medical Center, Seoul 06351, Korea; chchoi93@gmail.com (C.C.); onlyshohow@naver.com (A.S.); camuserik@gmail.com (S.-W.S.); 2Marine Natural Products Chemistry Laboratory, Korea Institute of Ocean Science and Technology, 385 Haeyangro, Busan 49111, Korea; yeonwoo@kiost.ac.kr; 3Department of Applied Ocean Science, University of Science and Technology, Daejeon 34113, Korea; 4Department of Radiation Oncology, Sungkyunkwan University School of Medicine, Seoul 06351, Korea

**Keywords:** (−)-agelamide D, radiation therapy, hepatocellular carcinoma, unfolded protein response (UPR), activating transcription factor 4 (ATF4)

## Abstract

Radiation therapy (RT) is an effective local treatment for unresectable hepatocellular carcinoma (HCC), but there are currently no predictive biomarkers to guide treatment decision for RT or adjuvant systemic drugs to be combined with RT for HCC patients. Previously, we reported that extracts of the marine sponge *Agelas* sp. may contain a natural radiosensitizer for HCC treatment. In this study, we isolated (−)-agelamide D from *Agelas* extract and investigated the mechanism underlying its radiosensitization. (−)-Agelamide D enhanced radiation sensitivity of Hep3B cells with decreased clonogenic survival and increased apoptotic cell death. Furthermore, (−)-agelamide D increased the expression of protein kinase RNA-like endoplasmic reticulum kinase/inositol-requiring enzyme 1α/activating transcription factor 4 (PERK/eIF2α/ATF4), a key pathway of the unfolded protein response (UPR) in multiple HCC cell lines, and augmented radiation-induced UPR signaling. In vivo xenograft experiments confirmed that (−)-agelamide D enhanced tumor growth inhibition by radiation without systemic toxicity. Immunohistochemistry results showed that (−)-agelamide D further increased radiation-induced ATF4 expression and apoptotic cell death, which was consistent with our in vitro finding. Collectively, our results provide preclinical evidence that the use of UPR inducers such as (−)-agelamide D may enhance the efficacy of RT in HCC management.

## 1. Introduction

Radiation therapy (RT) using ionizing radiation remains the mainstay treatment for solid tumors. Nonetheless, new therapeutic radiosensitizers that can potentially enhance tumor cell killing during RT are needed to improve treatment outcomes. Synthesized small molecules that increase the effect of RT have been discovered either by investigating new molecular entities or by repurposing known drugs [1,2]. There are several types of radiosensitizers developed for medical use, including nitroimidazoles capable of inducing radical-mediated DNA damage, N-oxide prodrugs that can be activated under hypoxic conditions [3], and thymine analogs that can be incorporated into DNA. Chemicals that affect the signal transduction pathways related to apoptosis or DNA repair have also been shown to increase the efficacy of RT. In addition, therapeutic antibodies such as cetuximab and nivolumab that block epidermal growth factor receptor (EGFR) signaling and immune checkpoints, respectively, have shown beneficial effects when combined with RT [4,5,6].

Efforts have been made to discover potent radiosensitizers from natural sources. Plant natural products that are capable of inhibiting or retarding the initiation of cancer by interacting with various cellular proteins have been considered as attractive candidate radiosensitizers [7]. Compounds such as resveratrol [8], genistein [9], curcumin [10], and quercetin [11] have been proven to enhance the tumoricidal effect of radiotherapy in preclinical settings, although their mechanism may vary. Based on these promising results, clinical studies have been conducted. A phase II clinical trial investigated the role of curcumin as a radiosensitizer for prostate cancer patients (NCT02724618) and a phase I/II trial investigated the effect of genistein supplements in RT for relieving pain caused by bone metastasis (NCT00769990).

Marine organisms have been exploited as promising sources of lead compounds for biomedical applications, but they are relatively unexplored for the development of radiosensitizers. Only a few marine natural products such as psammaplin [12], cephalosporine [13], and fucoidans [14] have been investigated for their activity to sensitize cancer cells to radiation. In our search for potent natural radiosensitizers from marine sources, we previously screened the radiosensitizing activity of various sponge extracts in human hepatocellular carcinoma (HCC) Hep3B cells and reported that the extracts obtained from *Agelas* sp. had radiosensitizing activity [15]. Our results clearly indicated that the *Agelas* extracts augmented the apoptosis of Hep3B cells by elevating endoplasmic reticulum (ER) stress during radiotherapy. Subsequent studies have focused on the identification of active pharmaceutical ingredients contained in the *Agelas* sponges (Appendix A), which led to the discovery of a potent natural radiosensitizer reported herein.

## 2. Results

### 2.1. Separation and Identification of Active Ingredients

Our previous study indicated that the extract from *Agelas* sp. contained ingredient(s) having a radiosensitizing activity for HCC [15]. Given that the radiosensitization may be related to ER stress, the expression levels of activating transcription factor 4 (ATF4) and microtubule-associated proteins 1A/1B light chain 3B (LC3B) were used as indicators to screen for radiosensitizing activity; ATF4 is a master transcription factor in the unfolded protein response (UPR), and LC3B is a key autophagy-related protein in Hep3B cells. For bioassay-guided isolation of compounds with radiosensitizing activity, the levels of these proteins induced by the fraction obtained from the extract were examined (Appendix A). After solvent partitioning and reverse-phase flash column chromatography of the crude extract (Appendix A), the resulting five fractions (50%, 30%, 10% aqueous methanol; methanol; and acetone fraction) were examined for their activity to modulate the levels of ATF4 and LC3B expression. The methanol fraction led to the strongest induction of the ATF4 and LC3B levels. This fraction was separated into seven subfractions through MPLC (medium-pressure liquid chromatography) using an octadecyl-silica (ODS) column, and each fraction was examined for its effect on the expression of ATF4 and LC3B. The fifth fraction induced ATF4 and LC3B expression (S2C) and significantly enhanced apoptotic cell death when combined with 4 Gy of X-rays (*p* < 0.001; Appendix A). This fraction was separated further into six subfractions through size exclusion column chromatography, and the strongest induction of LC3B and ATF4 was observed with the second and third subfractions, respectively. The purification of the second and third subfractions through high-pressure liquid chromatography (HPLC) yielded compounds **1** and **2**, respectively.

The comparison of nuclear magnetic resonance spectroscopy (NMR) and optical rotation measurement data with those reported in the literature indicated that compound **1** was (−)-agelasine D (Figure 1) [16,17]. The NMR data obtained for compound **2** coincided with (−)-ageloxime D, which was originally proposed as an oxime derivative [17] and revised later as a formamide derivative of agelasine D [18]. The molecular formula of C_26_H_40_N_5_O obtained by high-resolution mass spectrometry (HRMS) analysis was also consistent with the revised structure. According to the revised structure, the compound **2** was renamed (−)-agelamide D to represent the structure clearly. As detailed spectroscopic data of **2** were not provided in previous literatures, thorough 2D NMR analyses including homonuclear correlation spectroscopy (COSY), heteronuclear multiple bond correlation spectroscopy (HMBC), and nuclear Overhauser spectroscopy (NOESY) were performed in methanol-*d*_4_.

A group of ^1^H and ^13^C signals (positions 9, 11-17, 2′-6′, NCHO) for **2** appeared as two separate peaks (Table 1, Appendix A). Approaching the formamide groups increased the shift differences between two signals from the same position until the largest shift differences at C-15 (46.0, 41.6; Δ4.4) and C-5′ (97.3, 99.2; Δ1.9) attached directly to the amide nitrogen. It was speculated in the previous literature that the separated signals came from two rotamers resulting from the restricted rotation around the amide bond in the case of (+)-ageloxime D [18], although solid scientific evidence has not been provided. This speculation is convincing, as rotational isomers are often observed for tertiary amides synthesized [19,20] or isolated from natural sources [21,22]. As expected, COSY and HMBC correlations were clearly observed for each signal (Table 1, Appendix A), however, the hypothesis that the separate signals come from the rotamers could not be proved by spectroscopic evidence due to the lack of conclusive NOESY signals.

An optical rotation measurement of **2** yielded a value of −10.0 ± 0.1 (c 1.0, MeOH) at 25 °C, while the reported value of (−)-ageloxime D in the literature was −6.4 ± 0.6 (c 0.5, MeOH) [17]. In addition, the value reported for synthesized (+)-ageloxime D was +5.6 (c 0.5, MeOH) [18], which confirms the absolute stereochemistry of compound **2** as depicted in Figure 1.

### 2.2. (−)-Agelamide D Exerts a Radiosensitizing Effect on Hep3B Cells

Compound **1**, identified as (−)-agelasine D, was more cytotoxic to Hep3B cells than compound **2**, (−)-agelamide D (GI_50_ = 9.9 µM versus 12.0 µM; Figure 2A). Colony formation of Hep3B cells was inhibited more strongly by (−)-agelasine D than by (−)-agelamide D (Figure 2B,C). Western blotting showed that (−)-agelasine D induced LC3B expression to a greater extent than (−)-agelamide D, which corresponded to the results obtained with their mother fractions (Figure 2D). For ATF4, (−)-agelasine D and (−)-agelamide D showed similar level of induction. Based on these observations, (−)-agelamide D was chosen for further experiments on radiosensitization efficacy. The clonogenic survival assay revealed that (−)-agelamide D potentiated radiation-induced clonogenic death (Figure 2E,F). Furthermore, (−)-agelamide D augmented radiation-induced apoptotic cell death (Figure 2G,H). (−)-Agelamide D alone did not affect the cleavage of PARP (poly (ADP-ribose) polymerase), a surrogate marker of apoptosis, but its combination with 6 Gy of X-rays further increased the expression of cleaved PARP, relative to that by radiation alone (Figure 2G). Flow cytometry with annexin V staining confirmed that cotreatment with (−)-agelamide D and radiation resulted in higher induction of apoptosis (12.7%) than that by radiation alone (8.2%, *p* < 0.001, Figure 2H). These data suggest that (−)-agelamide D may enhance radiation sensitivity through apoptosis induction.

### 2.3. (−)-Agelamide D Augments Radiation-Induced ER Stress

To understand how (−)-agelamide D exerts the radiosensitizing effect, we first compared basal expression levels of UPR proteins in a panel of HCC cell lines, including Hep3B, Huh7, SNU-449, and PLC/PRF/5 (Figure 3A). SNU-449 cells showed the highest expression of protein kinase RNA-like endoplasmic reticulum kinase (PERK) and inositol-requiring enzyme 1α (IRE1α), even though the total protein levels were relatively low. In contrast, ATF4 expression was not detectable in SNU-449 cells. LC3B, a marker for autophagy, was the most abundant in Huh7 cells. Next, we tested how the expression of UPR proteins might be regulated by (−)-agelamide D in the four different HCC cell lines. In Hep3B cells, total PERK and IRE1α, as well as ATF4, increased by (−)-agelamide D in a concentration-dependent manner (Figure 3B). Huh7 cells showed a similar pattern, but to a lesser extent than that in Hep3B cells. In contrast, in SNU-449 cells, the expression of UPR proteins including ATF4 was not induced by (−)-agelamide D even at 5 μg/mL. PLC/PRF/5 cells also showed little induction of UPR proteins by (−)-agelamide D, except ATF4. SNU-449 cells showed slight induction of ATF4 after (−)-agelamide D treatment of up to 10 μg/mL (data not shown). These data indicated that HCC cell lines showed different responses to (−)-agelamide D due to diverse expression of UPR proteins.

To determine the combination effect of (−)-agelamide D and radiation, Hep3B cells were pretreated with 2 μg/mL of (−)-agelamide D for 3 h and then exposed to 6 Gy of X-rays. After 48 h, the radiation increased the expression of PERK, phospho-PERK, IRE1α, eukaryotic translation initiation factor 2 α subunit (eIF2α), and ATF4, which was greatly induced by the combined treatment with (−)-agelamide D (Figure 3C). In Huh7 cells, (−)-agelamide D was used at 5 μg/mL and increased expression of PERK, phospho-PERK, phospho-eIF2α, and ATF4 when combined with 6 Gy of X-ray (Figure 3D). These results indicated that (−)-agelamide D augmented the radiation-induced UPR signaling in HCC cells.

### 2.4. (−)-Agelamide-D-Induced Radiosensitization Mediated by the PERK/ATF4 Axis

We further investigated the mechanism underlying the radiosensitization by (−)-agelamide D. Treatment with 2 μM tunicamycin, a well-known ER stress inducer, increased the expression of ATF4 and LC3B, which was reversed by the pretreatment with GSK2656157, a selective PERK inhibitor (PERKi) in a concentration-dependent manner (Figure 4A). Similarly, the treatment with 2 μg/mL (−)-agelamide D increased phosphorylation of PERK, which was also suppressed by the pretreatment with PERKi (Figure 4B). Furthermore, the pretreatment with PERKi dose-dependently blocked (−)-agelamide D-activated PERK signaling, such as increased expression of total PERK, phospho-eIF2α, ATF4, and C/EBP homologous protein (CHOP) (Figure 4C). These data suggested that (−)-agelamide D may exert its effect through modulating the PERK/eIF2α/ATF4 axis.

Radiation is known to elicit ER stress, which is consistent with our data (Figure 3C). PERKi decreased total PERK and IRE1α expression in the presence of 6 Gy of X-rays (Figure 4D). In contrast to (−)-agelamide D, the combination of radiation and PERKi increased the expression of ATF4 (Figure 4D). It also increased cleaved PARP expression with a decrease in Bcl-2 expression, suggesting the induction of apoptotic signaling (Figure 4D). The induction of LC3B by either (−)-agelamide D or radiation was suppressed by PERKi (Figure 4C,D). ATF4 was dose-dependently increased by radiation in the Hep3B cells transfected with control siRNA, whereas it was completely suppressed in the cells transfected with ATF4 siRNA (Figure 4E). Depletion of ATF4 using siRNA enhanced radiation-induced cleavage of PARP but not the expression of LC3B (Figure 4E). Radiation-induced DNA damage signaling was also enhanced by ATF4 depletion, as evidenced by the increased phosphorylation of DNA-PKcs and ATM (Figure 4E). These results suggest that UPR activation and apoptosis regulated by radiation may be different from those by (−)-agelamide D.

### 2.5. (−)-Agelamide D Enhances the Efficacy of Radiotherapy in an HCC Xenograft Mouse Model

To determine the in vivo efficacy of (−)-agelamide D, we developed a Hep3B xenograft model by subcutaneously implanting Hep3B cells into athymic BALB/c nude mice. Tumor-bearing mice were randomized into four groups: (i) sham treatment, (ii) (−)-agelamide D alone (1.25 mg/kg/day, three times per week), (iii) X-ray irradiation alone (RT; three daily fractions of 3 Gy), and (iv) (−)-agelamide D plus X-rays (Figure 5A). No significant body weight loss was observed in mice during treatments (Figure 5B). On day 21, the growth of Hep3B xenograft tumors that received RT was significantly lower than that in the sham treatment (1086.0 ± 223.0 mm^3^ and 3165.2 ± 396.3 mm^3^, respectively; *p* < 0.001; Figure 5C,D). (−)-Agelamide D alone did not affect tumor growth (3273.3 ± 108.3 mm^3^), but its combination with RT resulted in higher inhibition of tumor growth (397.3 ± 60.3 mm^3^) than that by RT alone (*p* < 0.01; Figure 5C,D). Furthermore, blood samples were collected from mice that received RT or (−)-agelamide D plus RT for biochemical analysis. Biochemical parameters revealed that the addition of (−)-agelamide D had little effect on the metabolic profile in blood serum, although the creatine phosphokinase level in the serum was higher in mice that received RT plus (−)-agelamide D than in mice that received RT alone (*p* = 0.054, Appendix A).

Based on our in vitro results indicating that (−)-agelamide D induced activation of UPR signaling in Hep3B cells, we performed immunohistochemistry (IHC) staining of tumor tissues harvested on day 21. ATF expression was induced in the tumor tissue by either (−)-agelamide D or RT alone (*p* < 0.001; Figure 6A,B). The combined treatment with (−)-agelamide D and RT further enhanced ATF4 expression (*p* < 0.001; Figure 6A,B), which was consistent with in vitro results (Figure 3C). Transferase dUTP nick end labeling (TUNEL) staining of the tumor tissues from mice that received each treatment showed that RT but not (−)-agelamide D alone increased apoptotic cell numbers (*p* < 0.001; Figure 6A,C). The combination of (−)-agelamide D and RT greatly increased the number of apoptotic cells in the tumor tissues (*p* < 0.001). These results suggest that (−)-agelamide D enhanced the efficacy of RT in vivo, possibly through ATF4-mediated cell death.

## 3. Discussion

Agelasine is a group of compounds with a characteristic structure of adenine–diterpenoid conjugate frequently isolated from the marine sponge *Agelas*. In the imidazole portion of the adenine ring, one of the nitrogens is methylated and the other is attached to the diterpenoid, which is mostly bicyclic [23]. Agelasine D has the labdane diterpenoid and a variety of biological properties including antibacterial activity [17], cytotoxicity [4,17], and antiprotozoal activity [24]. Ageloxime D was named after agelasine D, as it was misidentified as an oxime derivative when discovered from the sponge *Agelas nakamurai*. As aforementioned, the structure was later revised to formamide [18], thus renamed agelamide. The growth inhibitory activity of ageloxime D against murine lymphoma cells, L5178Y (GI_50_, 12.5 μM), is approximately one-third of that by agelasine D (GI_50_, 4.0 μM) [17], which is consistent with our observations. To the best of our knowledge, the radiosensitizing activity or the mode of cytotoxicity has never been reported previously for agelasines and related derivatives.

Numerous studies have shown that cancerous cells activate UPR signaling as a prosurvival mechanism, which is then used to adapt to the harsh microenvironment such as hypoxia and acidic pH [25,26]. UPR signaling is governed by three sensor proteins: PERK, IRE1α, and activating transcription factor 6 (ATF6). Upon ER stress, glucose-regulated protein 78 (GRP78), a molecular chaperone for misfolded proteins, is released from the sensor proteins, triggering UPR activation. To restore ER function and sustain survival, UPR responses attenuate protein synthesis by inhibiting translation through PERK-dependent phosphorylation of (eIF2α) and IRE1α-dependent mRNA decay. Sustained ER stress leads to apoptosis as a death signal. Activated UPR signaling alters sensitivity of cancer cells to chemotherapy and radiotherapy, for which researchers exploit UPR signaling as a promising therapeutic target.

Ionizing radiation such as therapeutic X-rays induces oxidative damage leading to UPR responses [27]. It is still uncertain which of the UPR pathways should be targeted to gain benefit from RT. In oropharyngeal cancer, overexpression of GRP78, a master regulator of UPR signaling, is associated with a poor prognosis, and silencing of GRP78 abrogates radioresistance through inhibiting radiation-induced DNA repair [28]. Similarly, in preclinical settings, targeting GRP78 using antibodies enhances the efficacy of RT in glioblastoma, non-small-cell lung cancer, and pancreatic cancer [29,30]. Another study showed that the activation of ATF6 in response to RT contributes to RT-induced GRP78 upregulation, and knockdown of ATF6 is sufficient to enhance RT-induced cell death in glioblastoma [31]. Inversely, induction of ER stress by 2-deoxy-d-glucose or activation of UPR by inhibiting protein disulfide isomerase sensitizes glioblastoma cells to RT [32,33]. There are conflicting results regarding PERK inhibition: one study shows an increase in radioresistance by PERK knockdown [34], while another shows an increase in radiosensitization by PERK/ATF4/LAMP3 knockdown or treatment with PERK inhibitor [35]. The decision of cell fate between survival and death under ER stress depends on a difference in the amplitude of the intrinsic UPR signaling among cell types [25]. Our screening of a panel of HCC cell lines showed that the expression of UPR components varied across the cell lines. The PERK/eIF2α/ATF4 pathway is key for maintaining ER homeostasis in the tumor microenvironment. ATF4, a stress-induced transcription factor downstream of PERK, typically induces adaptive prosurvival response, but under excessive and persistent stress conditions, it promotes apoptosis via CHOP [36]. Our findings showed that (−)-agelamide D activated the PERK/ATF4/CHOP pathway, which was suppressed by PERKi, suggesting a possible role of the PERK/ATF4/CHOP pathway in (−)-agelamide D-mediated radiosensitization. However, the PERK inhibitor increased the expression of ATF4 and cleaved PARP in the presence of radiation, suggesting that radiation may activate ATF4/CHOP signaling through a PERK-independent route. Furthermore, the depletion of ATF4 in Hep3B cells increased the radiation-induced DNA damage and apoptotic death relative to that in the control siRNA treatment, suggesting a dual role of ATF4 in radiation-induced ER stress, switching between survival and death (Figure 7). Collectively, these results suggested that the enhanced ER stress by the combination of (−)-agelamide D and radiation might overwhelm the adaptive UPR, leading to an increase in apoptotic cell death. Consistently, the administration of (−)-agelamide D alone did not affect xenograft tumor growth in nude mice but enhanced tumor growth inhibition in combination with RT, which correlated with an increase in ATF4 expression and apoptotic cells within tumors.

Hepatic ER stress and UPR activation is associated with various liver diseases, such as nonalcoholic fatty liver disease and HCC [37]. ATF6, XBP1, and GRP78 increase in HCC tissues with advanced histological grading, suggesting the involvement of ER stress pathway in hepatocarcinogenesis [38]. In HCC cells, GRP78 is associated with an inferior response to sorafenib [39], the first targeted therapeutic drug approved for systemic treatment of advanced HCC. CHOP mediates ER stress-induced apoptosis in HCC cells [40]. Overexpression of ATF4 in HCC, but not in normal liver tissues, increases the resistance of HCC cells to chemotherapeutics [41]. RT is one of the effective local treatment options for unresectable HCC [42], but currently, there are no predictive biomarkers to guide treatment decision or adjuvant systemic drugs in RT for HCC. Thus, our results provide preclinical evidence that the activation of UPR by combined (−)-agelamide D and RT may be a promising strategy to manage advanced HCC.

## 4. Materials and Methods

### 4.1. Isolation and Structure Elucidation of Active Compounds from Agelas *sp.*

The specimens of *Agelas* sp. used for this study are the same ones used in our previous report [15]. Detailed procedures for sponge collection, extraction, and solvent partitioning are as follows. The sponge *Agelas* sp. was collected by hand using scuba at a 10-m depth offshore of Chuuk, Federated States of Micronesia, on 18 February 2010 (Appendix A). The voucher specimens were deposited at the Korea Institute of Ocean Science and Technology (registry No. 102CH-501). The lyophilized sponge (228.0 g) was extracted with methanol (1 L × 2) and dichloromethane (1 L × 1) at room temperature (Appendix A). The combined extract (26.2 g) was partitioned between n-butanol and water, and the organic layer (12.1 g) was partitioned again between 15% aqueous methanol and *n*-hexane. The aqueous methanol fraction (8.3 g) was then subjected to reversed-phase flash column chromatography (YMC Gel ODS-A, 60 Å, 230 mesh; YMC, Kyoto, Japan) with a stepped gradient elution of 50%, 30%, 10% aqueous methanol, methanol, and acetone. The methanol fraction (643.1 mg) was subjected to further separation through medium-pressure liquid chromatography (MPLC) using an ODS column (Redisep^®^ Rf C18; Teledyne ISCO, Lincoln, NE, USA) with a gradient elution changing from 30% aqueous methanol to 100% methanol followed by size-exclusion column chromatography (Sephadex^®^ LH-20; Cytiva, Marlborough, MA, USA) using 15% aqueous methanol as mobile phase. The fractions expected to contain active ingredient(s) were purified by reverse-phase high-pressure liquid chromatography (HPLC; YMC-Pack Pro C18; YMC, Kyoto, Japan) to obtain compounds **1** (10.4 mg) and **2** (11.2 mg).

The structures of the compounds **1** and **2** were elucidated by nuclear magnetic resonance spectroscopy (NMR) and high-resolution mass spectrometry (HRMS). ^1^H, ^13^C, and 2D NMR spectra were recorded on an ASCEND^TM^ 600 (Bruker Biospin Gmbh, Rheinstetten, Germany). Chemical shifts are reported in ppm from tetramethylsilane with the solvent resonance as the internal references (methanol-*d_3_*, δ_H_ 3.31 ppm, δ_C_ 49.00 ppm). HRMS data and the purity of the compounds were acquired by a Nexera X2 ultra-high-pressure liquid chromatography (UHPLC) system (Shimadzu, Kyoto, Japan) with a Gemin C18 column (Phenomenex, Torrance, CA), coupled with a Triple TOF^®^ 5600 + system (SCIEX, Framingham, MA, USA) using 0.1% formic acid in aqueous acetonitrile as a mobile phase. Optical rotation was measured in methanol using an Autopol^®^ III S2 (Rudolph Research Analytical, Hackettstown, NJ, USA) and IR spectra were recorded on a FT/IR-4100 (JASCO Inc., Easton, MD, USA). Compound **1** was identified as (−)-agelasine D by comparing its ^1^H, ^13^C NMR, optical rotation and HRMS data with those reported previously [16]. Structure of **2** was elucidated by comprehensive spectroscopic and mass analysis.

(−)-Agelamide D (**2**): Yellow amorphous solid; [α]${}_{D}^{25}$ −10.0 ± 0.1 (c 1.0, MeOH); IR (KBr) ν_max_ 3339, 2929, 1668, 1597, 1466, 1406, 1032 cm^−1^; ^1^H and ^13^C NMR (methanol-*d*_3_, 600 and 150 MHz), see Table 1; (+)-HRESIMS *m/z* 440.3388 [M + H]^+^ (calculated for C_26_H_42_N_5_O, 440.3389); Purity > 99%.

### 4.2. Cell Proliferation Assay.

Human HCC Hep3B, Huh7, PLC/PRF-5, and SNU449 cells were purchased from the Korean Cell Line Bank (Seoul National University, Seoul, Korea, 2016) and cultured as previously described [43]. All cell lines were tested annually for Mycoplasma contamination and were authenticated through short tandem repeat (STR) profiling.

The proliferation of HCC cells was evaluated by the Cell Counting Kit-8 assay (CCK-8, Dojindo Laboratories, Kumamoto, Japan). Hep3B cells were seeded at 1 × 10^3^ cells/well into a 96-well plate and incubated with various concentrations of (−)-agelamide D or (−)-agelasine D for 72 h. The cells were incubated with CCK-8 solution for additional 2 h at 37 °C and the absorbance was monitored at 450 nm using a SpectraMax i3 microplate reader (Molecular Devices, Sunnyvale, CA, USA). The relative cell viability was calculated as a percentage of the dimethyl sulfoxide (DMSO)-treated control.

### 4.3. Irradiation Experiments

X-ray irradiation was performed as previously described [15]. Briefly, the cell dishes were placed under a 2 cm-thick solid-water phantom with a source surface distance of 100 cm and a field size of 30 × 30 cm. 6-MV X-rays were delivered to the cells at a dose rate of 3.96 Gy per min using a linear accelerator Varian Clinac 6EX machine (Varian Medical Systems, Palo Alto, CA, USA). The absolute dose was calibrated according to TG-51 and verified using Gafchromic film to 1% accuracy.

### 4.4. Clonogenic Assay

To measure radiosensitivity, a clonogenic assay was performed as previously described [15]. Briefly, Hep3B cells were pretreated with 2 µg/mL of (−)-agelamide D for 3 h and were subsequently irradiated with increasing doses of 0, 2, 4, and 6 Gy of X-rays. After incubation for 14 days, cells were stained with 1% crystal violet and the colonies consisting of 50 or more cells were considered viable and scored. The plating efficiency was calculated as the percentage of colonies from seeded cells, and the cell survival fraction at each irradiation dose was determined by dividing the plating efficiency of the irradiated cells by that of the sham-treated control. The survival curves were drawn using GraphPad Prism 8.4.2 (GraphPad Software, La Jolla, CA, USA) with the linear–quadratic model (SF = exp(-αD-βD^2^); SF, survival fraction; D, absorbed dose).

### 4.5. Apoptosis Assay

Apoptosis was assessed through flow cytometry after annexin V/propidium iodide (PI) staining. Hep3B cells were pretreated with 2 µg/mL of (−)-agelamide D for 3 h, followed by exposure to 6 Gy of X-rays. After 48 h incubation, cells were detached with trypsin, washed with phosphate-buffered saline (PBS, pH 7.4), and stained with annexin V-FITC (BD Pharmingen, San Diego, CA, USA) and 2 μg/mL PI in annexin V binding buffer (10 mM HEPES, pH 7.4, 140 mM NaCl, 2.5 mM CaCl_2_) for 15 min at 37 °C in the dark. The apoptotic cell population was analyzed using a BD FACSVerse flow cytometer (Becton-Dickinson, CA, USA) with the data acquisition software, BD FACSuite.

### 4.6. Western Blot Analysis

Cells were harvested and lysed in lysis buffer (20 mM Tris, pH 8.0, 137 mM NaCl, 10% glycerol, 1% nonidet P-40, 10 mM EDTA, 100 mM NaF, 1 mM phenylmethylsulfonyl fluoride, and 10 mg/mL leupeptin). After centrifugation at 13,000 rpm for 15 min, protein concentration in each supernatant was determined using the Bio-Rad protein assay reagent (Bio-Rad, Richmond, CA, USA). For western blotting, the same amount of proteins was separated by sodium dodecyl sulphate-polyacrylamide gel electrophoresis (SDS-PAGE) and transferred to nitrocellulose membranes (Bio-Rad). After blocking with 5% skim milk in PBS at 4 °C, the blots were probed with primary antibodies overnight. After incubation with secondary antibodies for 1 h, bands of interest were visualized with Amersham enhanced chemiluminescence detection reagents (GE healthcare, Piscataway, NJ, USA). Representative images from at least two independent experiments are shown. The relative band intensity was quantified by ImageJ 1.52 and was normalized to β-actin.

### 4.7. Animal Experiments

The animal experiments were conducted in accordance with all appropriate regulatory standards under a protocol reviewed and approved by the Institutional Animal Care and Use Committee of the Samsung Biomedical Research Institute (approval number: 20190611002). Six to seven-week-old male Balb/c nude mice were purchased from Orient Bio (Gapyeong, Republic of Korea). Hep3B cells (5 × 10^6^) were injected subcutaneously into the right hind leg of each mouse. Tumor dimensions were measured twice a week with a caliper, and the tumor volume was calculated according to the formula: volume = L × W2 × 1/2 (L, length in nm; W, width in nm). When the mean tumor volume reached 80–150 mm^3^, mice were randomized into four groups; (i) sham group (no radiation), (ii) (−)-agelamide D (1.25 mg/kg/day) group, (iii) radiation therapy (RT, 3 Gy/day x 3 fractions) group, and (iv) (−)-agelamide D + RT group. (−)-Agelamide D was intraperitoneally administered three times a week starting on the day of randomization. The mice were euthanized 21 days after irradiation.

### 4.8. TUNEL Assay and Immunohistochemistry

Apoptotic cells in tumor tissue sections were detected using the terminal deoxynucleotidyl transferase dUTP nick end labeling (TUNEL) assay as previously described [44]. Irradiated tumor tissues were fixed with 10% neutral buffered formalin (NBF, Sigma-Aldrich, St. Louis, MO, USA) for 4 h and embedded in paraffin. After deparaffinization, TUNEL staining was performed using the In Situ Cell Death Detection Kit (Roche Diagnostics, Mannheim, Germany).

Immunohistochemistry (IHC) was performed as previously described [44]. Briefly, the tumor sections were sliced into 4 μm-thickness, deparaffinized in xylene, rehydrated in graded alcohol, and washed with 0.01 M PBS, pH 7.4. After heat-induced epitope retrieval with citrate buffer (pH 6.0; Dako, Carpinteria, CA, USA) and blocking with a blocking solution (Dako, Carpinteria, CA, USA), the tissue sections were incubated with anti-ATF4 rabbit polyclonal antibody (1:100; Abcam, Cambridge, UK) at 4 °C overnight. After washing with PBS, the samples were incubated for 30 min at room temperature with horseradish peroxidase-conjugated secondary antibodies (Dako, Carpinteria, CA, USA), and the slices were incubated with 3,3′-diaminobenzidine substrate chromogen solution (DAB, Dako, Carpinteria, CA, USA) for 5 min. Images for TUNEL and IHC were captured using an Aperio ScanScope AT slide scanner (Leica Biosystems Inc. Buffalo Grove, IL, USA) and analyzed using ImageScope software 12.4.3 (Leica Biosystems).

### 4.9. Statistical Analysis

Data are presented as the mean ± standard deviation from more than two independent experiments. Statistical analyses were performed using GraphPad Prism 8.4.2. Statistical significance of differences among groups was calculated with one-way or two-way ANOVA with Sidak’s multiple comparison post-hoc test. All *p*-values <0.05 were considered statistically significant.

## Figures and Tables

**Figure 1 marinedrugs-18-00500-f001:**
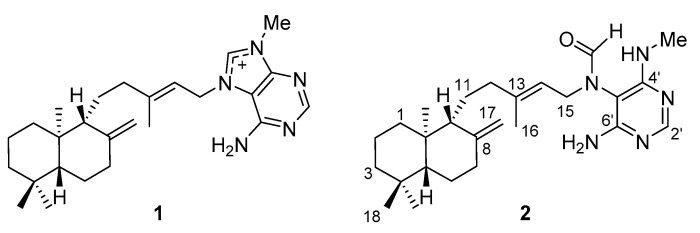
Structures of (−)-agelasine D (**1**) and (−)-agelamide D (**2**).

**Figure 2 marinedrugs-18-00500-f002:**
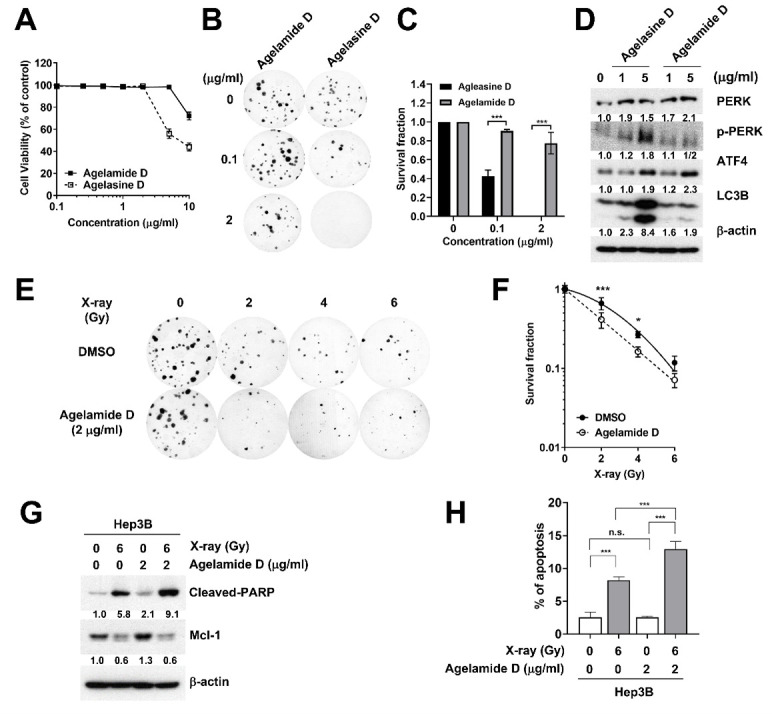
(−)-Agelamide D in vitro sensitizes Hep3B human hepatocellular carcinoma cells to radiation. (**A**) (−)-Agelamide D was less cytotoxic than (−)-agelasine D in Hep3B cells. The proliferation of Hep3B treated with various concentrations of (−)-agelamide D was evaluated using the Cell Counting Kit 8 (CCK8) assay. Data are presented as the mean ± standard deviation (SD) of two independent experiments (*n* = 6). (**B**) (−)-Agelamide D inhibited clonogenic survival to a lesser extent than (−)-agelasine D. **(C)** Quantification of survival fraction after treatment with (−)-agelamide D and (−)-agelasine D. Survival fraction was calculated as described in Materials and Methods. Data are presented as the mean ± SD of three independent experiments (*n* = 6). Difference was evaluated using two-way analysis of variance (ANOVA), followed by Sidak’s multiple comparisons test. *** *p* < 0.001. (**D**) (−)-Agelamide D induced ER (endoplasmic reticulum) stress and autophagy, but to a lesser extent than (−)-agelasine D. LC3B was included as an autophagy marker. (**E**) and (**F**) (−)-Agelamide D decreased clonogenic survival after irradiation. Hep3B cells were pretreated with 0.1 μg/mL of (−)-agelamide D for 3 h and then irradiated with the indicated doses of X-rays. Surviving colonies were stained with crystal violet and counted. Representative images (**E**) and dose response curves (**F**). Data are presented as the mean ± SD of three independent experiments (*n* = 9). * *p* < 0.05.; *** *p* < 0.001. Difference was evaluated using two-way ANOVA, followed by Sidak’s multiple comparisons test. (**G**) and (**H**) (−)-Agelamide D augmented radiation-induced apoptosis in Hep3B cells. (**G**) Western blot analysis showed that the pretreatment with (−)-agelamide D enhanced radiation-induced cleavage of PARP. β-Actin was used as a loading control. (**H**) Flow cytometry analysis with annexin V staining showed that (−)-agelamide D augmented radiation-induced apoptosis. The percentage of total apoptotic cells (early and late apoptotic) was quantified. Data are presented as the mean ± standard deviation (SD) of three independent experiments (*n* = 3). Difference was evaluated using one-way ANOVA, followed by Tukey’s multiple comparisons test; *** *p* < 0.001; n.s., not significant.

**Figure 3 marinedrugs-18-00500-f003:**
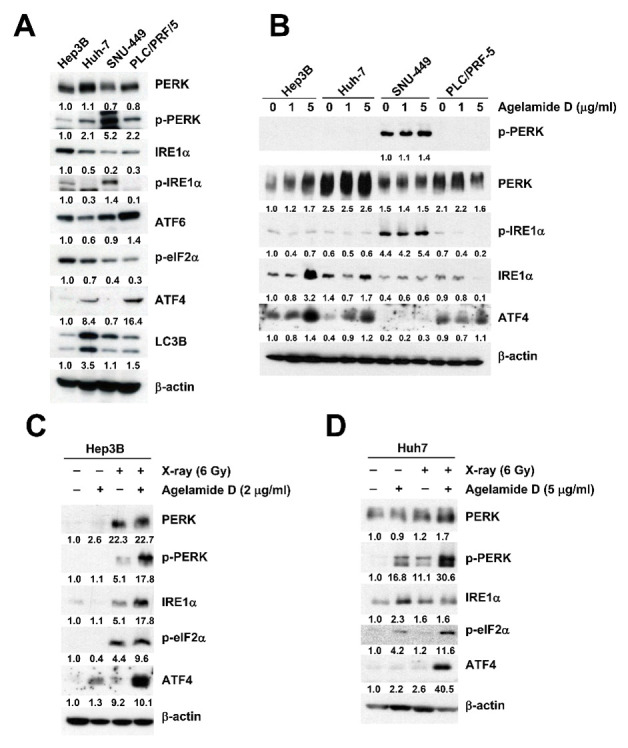
(−)-Agelamide D augments radiation-induced activation of UPR signaling in HCC cell lines. (**A**) Basal expression levels of UPR pathway proteins in four different HCC cell lines: Hep3B, Huh7, SNU-449, and PLC/PRF/5. SNU449 showed strong activation of PERK and IRE1α while Huh7 showed higher LC3B expression. (**B**) Comparison of UPR activation by (−)-agelamide D in four different HCC cell lines. (−)-Agelamide D increased ATF4 expression in Hep3B, Huh7, and PLC/PRF/5 but not in SNU449 cells. (**C**) Combined treatment with (−)-agelamide D and radiation further increased PERK and ATF4 in Hep3B cells. (**D**) Combination of (−)-agelamide D and radiation activated PERK/eIF2α/ATF axis in Huh7 cells.

**Figure 4 marinedrugs-18-00500-f004:**
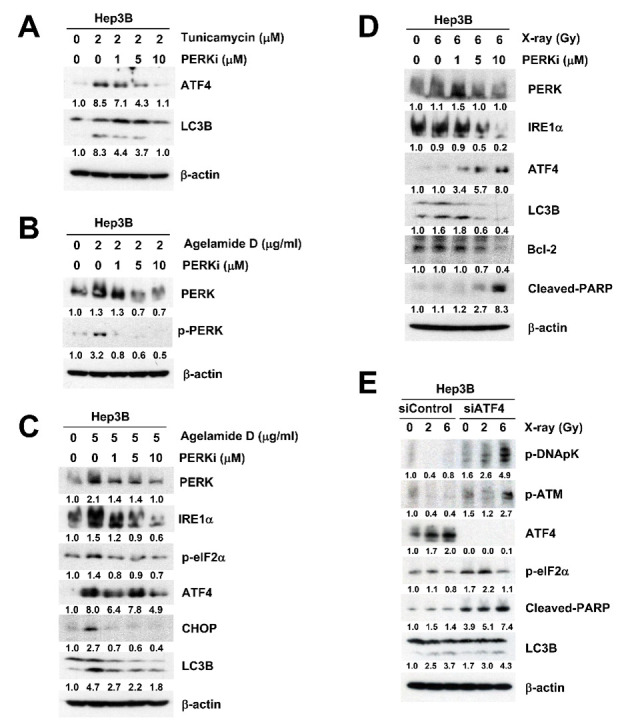
(−)-Agelamide D exerts its radiosensitization via PERK/eIF2α/ATF4. (**A**) GSK2656157, a selective PERKi, suppressed the induction of ATF4 and LC3B by tunicamycin in a concentration-dependent manner. Hep3B cells were pretreated with indicated concentrations of GSK2656157 for 1 h, followed by incubation with 2 μM tunicamycin. After 24 h, the cells were collected for western blotting. (**B**) PERKi dose-dependently inhibited (−)-agelamide D-induced PERK activation. (**C**) PERKi inhibited (−)-agelamide D-mediated activation of the PERK/eIF2α/ATF4/CHOP axis. (**D**) PERKi further increased radiation-induced expression of ATF4 and cleaved PARP. Hep3B cells were treated with PERKi 1 h prior to irradiation with 6 Gy of X-rays and then incubated for 24 h. (**E**) ATF4 knockdown using siRNA increased radiation-induced cleaved PARP expression. β-actin was used as a loading control.

**Figure 5 marinedrugs-18-00500-f005:**
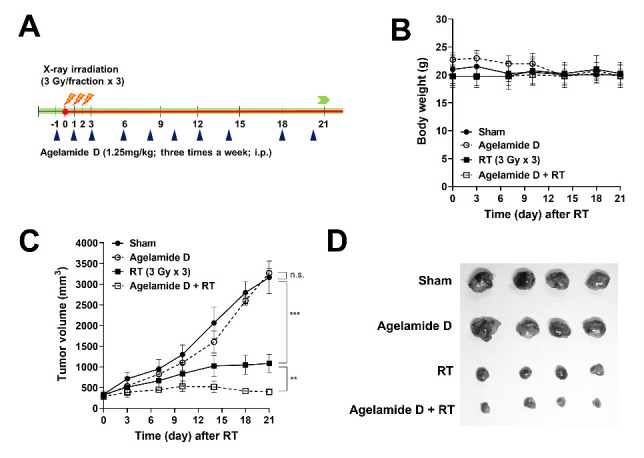
(−)-Agelamide D enhances radiation-mediated tumor growth inhibition in a Hep3B xenograft mouse model. (**A**) Schematic diagram of the experimental procedure. The Hep3B tumor-bearing mice were randomized into four groups: (i) sham group, (ii) (−)-agelamide D alone (1.25 mg/kg/day), (iii) radiation therapy (RT, 3 Gy/day × 3 fraction) group, and (iv) (−)-agelamide D + RT group. (**B**) Body weight change during the treatment. No significant difference in body weight was observed among groups. (**C**) Growth curves of Hep3B xenograft tumors in nude mice. Hep3B cells were implanted into right legs of BALB/c nude mice. Once tumors were palpable, mice were intraperitoneally injected with (−)-agelamide D three times a week. The tumors were irradiated with 3 Gy of X-ray for 3 consecutive days for a total 9 Gy. Mean tumor volumes and their standard deviation per group (*n* = 4). Statistical significance was determined by two-tailed paired *t*-test. ** *p* < 0.01; *** *p* < 0.001. (**D**) Photographs of tumors harvested from mice 21 days postirradiation.

**Figure 6 marinedrugs-18-00500-f006:**
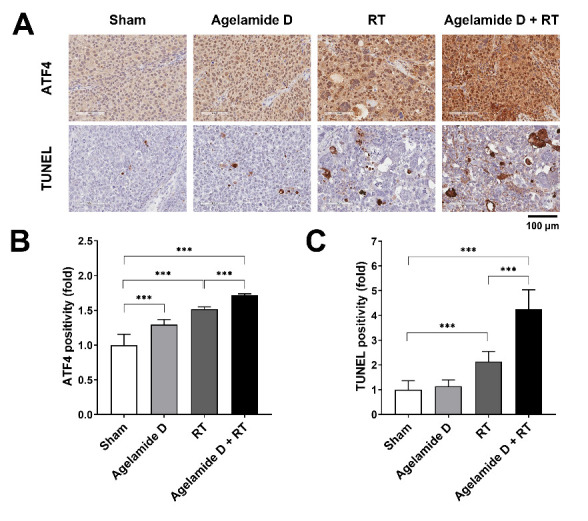
(−)-Agelamide D enhances radiation-induced expression of ATF4 and apoptotic cell death in vivo. (**A**) Representative images of immunohistochemistry of xenograft tumor tissues. UPR activation and apoptotic cell death were assessed through ATF4 and TUNEL positivity, respectively. (**B**) and (**C**) Quantification of ATF4 (**B**) and TUNEL (**C**) positivity in Hep3B xenograft tumor tissues. Tumors from each treatment group were harvested after 21 days of irradiation and then embedded in paraffin blocks. Immunohistochemistry was performed as described in the Materials and Methods. Data are presented as the mean ± SD (*n* = 15). Statistical significance was determined by one-way ANOVA, followed by Tukey’s multiple comparisons test. *** *p* < 0.001.

**Figure 7 marinedrugs-18-00500-f007:**
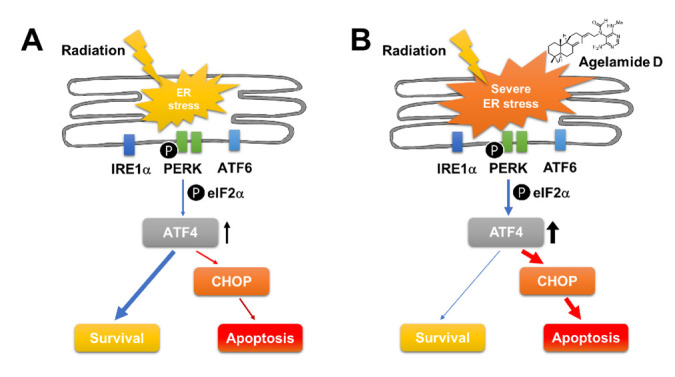
Schematic diagram illustrating the proposed mechanism of (−)-agelamide D-mediated radiosensitization in HCC. (**A**) Ionizing radiation induces ER stress via reactive oxygen species (ROS) generation, which triggers unfolded protein response. Activation of PERK/eIF2α/ATF4 pathway promotes survival, leading to radioresistance. (**B**) Combination with radiation and algeloxime D induces severe ER stress. Elevated level of ATF4 switches from adaptive survival to apoptotic cell death via CHOP-mediated apoptotic signaling.

**Table 1 marinedrugs-18-00500-t001:** Nuclear magnetic resonance spectroscopy (NMR) spectroscopic data (600 MHz, methanol-*d*_4_) for compound **2.**

Position	δ_C_ *^a^*	δ_H_ (*J* in Hz)	COSY	HMBC
1a	40.3	0.98, m	1.61	-
1b		1.51, m	-	15.1
2a	20.4	1.49, m	-	-
2b		1.61, ddd (13.3, 3.2, 3.2)	0.98	-
3a	43.3	1.22, m	1.40	-
3b		1.40, brd (13.3)	1.22	56.4, 40.7
4	40.7	-	-	-
5	56.4	1.11, dd (12.6, 3.0)	1.31	40.7, 57.7, 34.5, 15.1
6a	25.6	1.31, m	1.74, 2.37, 1.11, 1.96	-
6b		1.74, m	-	56.4, 149.8
7a	39.4	1.96, m	2.37, 1.31	149.8, 106.9, 25.6
7b		2.37, m	1.96, 1.31	149.8, 106.9, 56.4, 25.6
8	149.8	-	-	-
9	57.8/57.7	1.52, m/1.52, m	4.81, 4.46, 4.44	23.0, 106.9, 149.8
10	34.5	-	-	-
11a	23.0/23.1	1.52, m	2.05	-
11b		1.29, m	-	
12a	39.6/39.7	2.05, m	1.52, 4.14	16.0, 23.0, 118.0, 144.3
12b		1.74, m	-	144.4, 118.0
13	144.4/144.3	-	-	-
14	118.4/118.0	5.29, t (7.8)/5.21, t (7.8)	4.14, 2.05/4.23	16.0, 39.6, 46.0/16.2, 39.7
15	46.0/41.6	4.14, m/4.23, m	5.29 5.21	97.3, 118.4,144.4, 165.9/99.2, 118.0,144.3, 166.5
16	16.0/16.2	1.49, s/1.56, s	-	39.6, 118.4, 144.4/39.7, 118.0, 144.3
17a	106.9/106.9	4.81,	1.52	57.8, 39.4
17b		4.46, brs/4.44, brs	1.52/1.52	39.4, 57.8, 149.8
18	34.1	0.89, s	-	56.4, 43.3, 40.7, 22.2
19	22.2	0.82, s	-	56.4, 43.3, 34.1
20	15.1	0.69	-	57.7, 40.3,
2′	157.9/157.5	7.92, s/7.92, s	-	161.6
4′	162.0/161.6	-	-	-
5′	97.3/99.2	-	-	-
6′	160.6/160.0	-	-	-
NCHO	165.9/166.5	8.20, s/7.91, s	-	97.3, 46.0 /99.2, 41.6
NMe	28.2	2.88, s	-	161.6

^a^ Carbons correlating with the corresponding pµroton.

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
