# Peer review of "Therapeutic Potential of (−)-Agelamide D, a Diterpene Alkaloid from the Marine Sponge Agelas sp., as a Natural Radiosensitizer in Hepatocellular Carcinoma Models"

_marinedrugs, 2020, doi:10.3390/md18100500_

Round 1
Reviewer 1 Report
In this study, the authors identified compounds with therapeutic potential for HCC from marine sponge Agela sp. The authors separated and identified active ingredients, and found Ageloxime D is one of their interested compound. The authors then studied potential roles of Ageloxime D on HCC cell lines and xenograft tumors. Mechanically, Ageloxime D may cause ER-stress and enhance the sensitivity of tumor cells to radiation. Overall, the study is interesting and systematic.
Following are a few points the authors may consider to further improve the manuscript.
1. Figure 2E. What is the survival fraction?
2. How many times were the in vitro experiments repeated?
3. The authors should quantify and show all immunoblotting results instead of only showing representative blots.
Reviewer 2 Report
The manuscript submitted by Choi et al. shows the isolation and the bioactive potential of ageloxime D, a marine sponge derived compound, for the treatment of HCC in combination with RT. The characterization of new tools to enhance conventional treatments of cancer is a very interesting line of research in pharmacology and critical for biomedicine, justifying the importance of this work in the field. The manuscript is very well written and data are correctly explained along the text. After the revision of the manuscript, I have some suggestions and comments for the authors, listed bellow:
- In line 69, reference 15 is not correctly formatted according to bibliography style.
- In the manuscript, the order of supplementary figures is not corrected: the first figure mentioned is FigS2B, figure S1 is mentioned in MM section, at the end of the manuscript…). Please modify the text and/or order of the supplementary figures to correctly mention the figures.
- In figure S2A, lines of the scheme should be darker in order to better visualization.
- Please increase the text size in Figure 5A
- Statistics in Figure 2E is not included
- Statistic and quantification in Figure 2B are not included.
- Please, indicate the value of IG50 calculated for ageloxime D and agelasine D in Hep3B cells (from data of Figure 2A).
- In MM, apoptosis assay with annexin V is described, but authors do not indicate the subpopulation of cells included in the quantification data (early or late apoptosis, or both).
- In Figure 3B, ageloxime D 5 ug/mL did not induce p-PERK in Huh-7 cells, but in Fig3D, the compound greatly increased p-PERK in these cells. Please explain the difference in the obtained results.
- The proposed mechanism of action of ageloxime D in HCC sensitization to RT should be included in a schematic figure.
- Please indicate the number of independent experiments performed in each case.
Reviewer 3 Report
I have examined the NMR part (line #86-#112 and related tables and figures) of the manuscript as an NMR expert upon request from Dr. Edrada-Ebel. NMR experiments seems to be performed and analyzed appropriately, but the results part is difficult to understand due to poor writing and a confusing figure.
Lines #87-#99:
Most data and analyses described in lines #87-#99 are from ref #18, but not clearly explained. The authors should clarify new findings and data. Structure "3" shown in Figure 1 is confusing (see below, about Figure 1).
Stereoisomers should be mentioned. At least, enantiomers should be distinguished.
Figure 1:
This figure is confusing. The structures of ageloxime D (2, 3) are from ref #17 and #18, and the structure "3" is not directly related to this manuscript. The structure of ageloxime D is not revised in this manuscript, but the revised structure is shown.
Lines #102-#112:
Double (twin) peaks are explained by rotamers. However, this molecule can have other stereoisomers, including enantiomers. A careful explanation is required.
Figures S5 and S6:
The resolution of the figure is too low to follow the assignments. No NOESY spectrum.
